# Clinical Snapshot of Group A Streptococcal Isolates from an Australian Tertiary Hospital

**DOI:** 10.3390/pathogens13110956

**Published:** 2024-11-01

**Authors:** Phoebe K. Shaw, Andrew J. Hayes, Maree Langton, Angela Berkhout, Keith Grimwood, Mark R. Davies, Mark J. Walker, Stephan Brouwer

**Affiliations:** 1Australian Infectious Diseases Research Centre, Institute for Molecular Bioscience, The University of Queensland, Brisbane, QLD 4067, Australia; phoebe.shaw@uq.edu.au (P.K.S.); mark.walker@uq.edu.au (M.J.W.); 2Department of Microbiology and Immunology, Peter Doherty Institute for Infection and Immunity, The University of Melbourne, Melbourne, VIC 3000, Australia; andrew.hayes@unimelb.edu.au (A.J.H.); mark.davies1@unimelb.edu.au (M.R.D.); 3Gold Coast Laboratory, Pathology Queensland, Gold Coast, QLD 4215, Australia; maree.langton@health.qld.gov.au; 4Infection Management and Prevention Service, Queensland Specialist Immunisation Service, Children’s Health Queensland, South Brisbane, QLD 4101, Australia; angela.berkhout@health.qld.gov.au; 5Queensland Statewide Antimicrobial Stewardship Program, Department of Paediatrics, Royal Brisbane and Women’s Hospital, Gold Coast University Hospital, Southport, QLD 4215, Australia; 6Gold Coast and Departments of Infectious Diseases and Paediatrics, Gold Coast Health, School of Medicine and Dentistry, Griffith University, Gold Coast, QLD 4215, Australia; k.grimwood@griffith.edu.au

**Keywords:** *Streptococcus* pyogenes, scarlet fever, invasive infection, *emm* types, superantigen, streptolysin O

## Abstract

*Streptococcus pyogenes* (Group A *Streptococcus*, GAS) is a human-restricted pathogen that causes a wide range of diseases from pharyngitis and scarlet fever to more severe, invasive infections such as necrotising fasciitis and streptococcal toxic shock syndrome. There has been a global increase in both scarlet fever and invasive infections during the COVID-19 post-pandemic period. The aim of this study was the molecular characterisation of 17 invasive and non-invasive clinical non-*emm*1 GAS isolates from an Australian tertiary hospital collected between 2021 and 2022. Whole genome sequencing revealed a total of nine different GAS *emm* types with the most prevalent being *emm*22, *emm*12 and *emm*3 (each 3/17, 18%). Most isolates (14/17, 82%) carried at least one superantigen gene associated with contemporary scarlet fever outbreaks, and the carriage of these toxin genes was non-*emm* type specific. Several mutations within key regulatory genes were identified across the different GAS isolates, which may be linked to an increased expression of several virulence factors. This study from a single Australian centre provides a snapshot of non-*emm*1 GAS clinical isolates that are multiclonal and linked with distinct epidemiological markers commonly observed in high-income settings. These findings highlight the need for continual surveillance to monitor genetic markers that may drive future outbreaks.

## 1. Introduction

*Streptococcus pyogenes*, also known as Group A *Streptococcus* (GAS), is a Gram positive, β-haemolytic, facultatively anaerobic bacterial pathogen. GAS is exquisitely adapted to the human host, causing a wide range of clinical manifestations, ranging from mild infections such as pharyngitis, scarlet fever, and impetigo to more severe invasive diseases including bacteraemia, necrotising fasciitis and streptococcal toxic shock syndrome [1]. Additionally, GAS infection can trigger post-infection autoimmune sequelae, such as acute rheumatic fever and poststreptococcal glomerulonephritis [2]. It is estimated that GAS causes approximately 500,000 deaths worldwide annually [3], predominantly from rheumatic heart disease where there is an inequitable burden impacting lower middle-income countries and Indigenous populations of affluent nations [4,5]. Quantifying the disease burden of GAS infections remains difficult due to under-reporting of both invasive and non-invasive cases, a lack of comprehensive disease registries and reliance on passive surveillance [6]. There is no commercial vaccine currently available against GAS, and the development of a safe and effective vaccine remains a high priority. GAS infections can be treated with antibiotics [7] with GAS remaining remarkably susceptible to β-lactams [8]. 

A global re-emergence of scarlet fever has been reported since 2011 [9,10,11]. In some cases, GAS pharyngeal infection may result in acute rheumatic fever and rheumatic heart disease with long-term damage to the heart valves [12,13]. In 2022, the World Health Organization (WHO) reported spikes in both scarlet fever and invasive GAS (iGAS) infections in several European countries, including the United Kingdom (UK), France, Ireland, Sweden, the Netherlands and Australia [4,14]. iGAS disease is defined as the isolation of strains from normally sterile body sites, such as blood, pleural and joint fluid or deep tissue [15,16]. Although progression from non-invasive to iGAS infection is rare, invasive infections are associated with mortality rates of up to 25%.

GAS isolates are classified into *emm* types based on the hypervariable 5′ sequence of the M protein, which is a surface-anchored protein and key virulence determinant [17]. Each GAS isolate harbours an *emm* gene variant with over 275 *emm* types documented currently [18]. Typing GAS isolates using *emm* sequencing remains important in the epidemiological surveillance of GAS infections [11]. Prevalent *emm* types vary across different geographical locations [19,20,21,22,23,24]. 

The concerning post-pandemic spike in both scarlet fever and iGAS cases, possibly due to a combination of events including the emergence of the M1_UK_ lineage, viral co-infections, and a potential decrease in the immune status of the general population following the relaxation of SARS-CoV-2 restrictions, has highlighted the requirement for the adequate detection and surveillance of GAS infections [25,26,27,28,29,30]. The presence of the M1_UK_ lineage has already been reported recently in Australia, and its dominance in the GAS population has been reported globally [27,31]. M1_UK_ isolates exhibit an increased production of superantigen SpeA, which may contribute to its fitness [27,31]. Contemporary scarlet fever isolates are also associated with a specific toxin repertoire, including superantigens SpeC and SSA, alongside deoxyribonuclease (DNase) Spd1 [1,20,23,32,33]. Therefore, we aimed to understand the prevalence of toxin repertoires in the population of non-*emm*1 clinical GAS isolates. Here, we performed *emm* typing and conducted a molecular analysis of 17 non-*emm*1 GAS clinical isolates from a tertiary hospital in southeast Queensland, Australia.

## 2. Materials and Methods

### 2.1. Bacterial Strains and Growth Conditions

A total of 17 non-*emm*1 clinical GAS isolates were collected from the Gold Coast University Hospital between 2021 and 2022 (Table 1). No patients succumbed to infection. Isolates were cultured on horse blood agar (HBA) or in Todd–Hewitt broth supplemented with 1% yeast extract (THY) and incubated at 37 °C.

### 2.2. Genome Sequencing, Assembly and Analysis

Genomic DNA (gDNA) was extracted from all isolates and used to create paired-end multiplex libraries, which were sequenced using the Illumina HiSeq 2500 platform at a length read of 150 (bp) (Australian Genome Research Facility, Brisbane, Australia). Read mapping was performed using Shovill V1.1.0 (https://github.com/tseemann/shovill, accessed on 14 May 2024) with an underlying SPAdes assembler [34]. Genome contigs were aligned in FASTA format with gene predictions and annotations generated using PROKKA [35] and streptococcal RefSeq-specific databases. Screening for virulence genes and regulators of interest was performed using screen_assembly v1.2.7 [36]. Methods were followed according to Davies et al. [36].

### 2.3. Phylogenetic Analysis

Mashtree [37] was used to place the 17 GAS isolates genomes in this study with a previously analysed 2083 global GAS genome sequences from Davies et al. [36].

### 2.4. Polymerase Chain Reaction (PCR) Screening

GAS toxin genes *slo* (streptolysin O), *speA* (streptococcal pyrogenic exotoxin A), *ssa* (streptococcal superantigen), *speC* (streptococcal pyrogenic exotoxin C), *spd1* (streptococcus pyogenes deoxyribonuclease 1), *speB* (streptococcal pyrogenic exotoxin B) and *hasA* (hyaluronan synthase), and macrolide-resistance gene *ermB* (erythromycin methylase), were each amplified by PCR using the KAPA HiFi PCR kit (Roche, cat no. 07958846001) and gDNA as a template. Primers and reagents parameters are listed in Appendix A. gDNA from SP1380 [31] served as a positive control for all toxins tested, and gDNA from HKU16 [23] served as a positive control for *ermB.* Distilled H_2_O (dH_2_O) was used as a negative control for all testing. PCR products were resolved on a 2% TAE agarose gel.

### 2.5. Sodium Dodecyl Sulfate–Polyacrylamide Electrophoresis (SDS-PAGE) and Western Blotting

Cultures were grown in THY supplemented with 2 mM cysteine to the late exponential phase to support SSA expression, as demonstrated previously [36]. Filter-sterilised culture supernatant was precipitated using 10% trichloroacetic acid (TCA). The precipitate was then resuspended in LDS loading buffer (NuPAGE) containing 100 mM dithiothreitol (DTT) normalised to the OD_600_ of the culture. Samples were boiled for 5 min, subjected to SDS-PAGE, and then transferred to a PVDF membrane (Sigma Aldrich, # IPFL00010, Saint Louis, MO, USA) by wet transfer for the detection of immune-reactive bands using a LI-COR Odyssey Imaging System (LI-COR Biosciences, Lincoln, NE, USA). The primary antibodies used to detect the SpeA, SpeC, SSA and SpeB protein in GAS culture supernatants were rabbit antibodies to SpeA (#PAI11, Toxin Technology, Sarasota, FL, USA), SpeC (PCI333, Toxin Technology), SSA (produced by Mimotopes [37]), and SpeB (PBI222, Toxin Technology) respectively, all at 1:1000 dilution. The murine primary antibodies to detect Spd1 and SLO protein in GAS culture supernatants were at 1:1000 dilution [36]. Of note, the Spd1 blot was incubated in primary antibody that had first been pre-adsorbed using HKU16Δ*ssa/speC/spd1* [36] to minimise non-specific bands. Briefly, HKU16Δ*ssa/speC/spd1* was subjected to SDS-PAGE, then transferred to a PVDF membrane via wet transfer as previously described. The membrane was then incubated in murine antibody to Spd1 at a 1:1000 dilution. This antibody then served as the primary antibody for the Spd1 blots. Either anti-rabbit IgG ((H + L) (DyLight™800 4X PEG Conjugate, NEB #5151P)) or anti-mouse IgG ((H + L) (Dylight™ 800 4X PEG Conjugate, NEB #5257S)) were used as secondary antibodies at a 1:10,000 dilution. Western blot images were captured using an Odyssey LI-COR instrument and Image Studio (v.5.2) software. Culture supernatants from HKU488 [31] served as a positive control for all toxins.

### 2.6. SpeB Caseinolytic Activity Assay

All isolates were grown on HBA plates overnight at 37 °C. GAS M1_global_ isolate 5448 was used as a positive control, while isolates 5448AP and 5448Δ*speB* [38] were used as negative controls. Cultures were statically grown to an OD_600_ of 0.4 in THY where 5 μL of undiluted culture was pipetted onto Columbia agar plus 15% skin milk and incubated at 37 °C for 72 h. SpeB activity was determined after 72 h based on the clear zone with an opaque perimeter surrounding the culture growth indicating caseinolytic activity [39].

### 2.7. SLO Activity Assay

SLO activity was measured by the standard SLO haemolysis assay [40]. All isolates were grown on HBA plates overnight at 37 °C. GAS M1_global_ isolate 5448 was used as a positive control, while isolate 5448Δ*slo* was used as a negative control. All bacterial cultures were statically grown in THY to late exponential phase with the exception of SP1494, which was grown in THY supplemented with 20 mM NaHCO_3_ to compensate for its in vitro growth defect characteristic of emergent, chimeric *emm*4 GAS [41]. Fresh whole human blood was centrifuged at 500 × *g* for 10 min to separate red blood cells (RBCs), buffy coat and plasma. The buffy coat and plasma were aspirated and discarded. The tube was filled to the original level of plasma with Hanks Balanced Salt Solution (HBSS), inverted gently to mix, and centrifuged at 500 × *g* for 10 min. The washing step was repeated for a total of two washes. The supernatant was aspirated and replaced with HBSS, inverted to mix, and was considered to contain only RBCs. A 2% RBC solution was prepared in HBSS, and 200 µL was aliquoted into a 96-well plate. Then, 40 µL of filter-sterilised culture supernatant was added to each well. Triton X-100 (final concentration of 1%) was used as a positive control for 100% RBC lysis, while phosphate-buffered saline was used as a negative control. The plate was incubated for 30 min at 37 °C and then centrifuged at 1000 × *g* for 10 min to pellet any intact RBCs. The amount of extracellular haemoglobin was measured spectrophotometrically at 405 nm using a CLARIOStar Plus microplate reader (BMG LABTECH). Results were graphed using GraphPad Prism version 10.2.0. Blood samples from a minimum of three healthy adult donors were used and performed in triplicate.

### 2.8. Ethics Approvals

Clinical approval for this project was granted by the Children’s Health Queensland Human Research Ethics Committee HREC/10/QRCH/113. The ethics approval number for the blood collected for the SLO activity assay is 2010/HE001586).

### 2.9. Statistical Analysis

All statistical analyses were completed using Prism software (GraphPad; version 10.2.0). Significance was calculated using one-way ANOVA and Tukey’s post-test.

### 2.10. Data Availability

Whole genome sequencing data obtained in this study were submitted to the Sequence Read Archive database (https://www.ncbi.nlm.nih.gov/sra, accessed on 30 September 2024; BioProject accession PRJNA1172375).

## 3. Results

### 3.1. Investigation of 17 Contemporary GAS Isolates

A total of 17 non-*emm*1 clinical GAS isolates were collected as a clinical snapshot from the Gold Coast University Hospital in subtropical southeast Queensland between 2021 and 2022. Of the 17 clinical isolates, 12 were isolated from invasive infections (iGAS) and 5 were non-invasive (scarlet fever [*n* = 4] and tonsillitis [*n* = 1]) (Table 1). A total of nine different *emm* types were identified following whole genome sequencing (WGS) (Table 1). *emm*22, *emm*12 and *emm*3.93 were the most common *emm* types (three isolates each), *emm*41 and *emm*4 had two isolates each, while *emm*89, *emm*77, *emm*75 and *emm*53 were each represented by a single isolate. Interestingly, (6/17, 35%) isolates were lacking the *hasABC* operon genes required for capsule biosynthesis (SP1494 and SP1508 (*emm*4), SP1501, SP1502 and SP1505 (*emm*22), and SP1503 (*emm*89)) [42]. A maximum-likelihood phylogenetic tree was constructed to understand the distribution of the 17 clinical GAS isolates across the global GAS population comprising a geographically and clinically diverse set of 2083 genomes with 150 different *emm* types [36]. All 17 isolates were found to be represented across the global GAS population (Figure 1a). All *emm*4, *emm*12 and *emm*41.2 isolates were found to be invasive. Of the five non-invasive isolates, three were acapsular.

### 3.2. Detection of Diverse Toxin Profiles in GAS Clinical Isolates

To study the toxin profile of the 17 clinical isolates, we performed PCR screening to confirm the presence of scarlet fever-associated superantigen toxins *speA*, *speC*, and *ssa* and the DNase gene *spd*1 [11,44] (Figure 1b). Of the superantigens, *ssa* was identified in 59% (10/17) of isolates, *speA* was detected in 7/17 isolates (42%), while *speC* and *spd1* were detected in 8/17 isolates (47%). Two of the three *emm*12 isolates (SP1492 and SP1493) were found to carry *ssa*, *speC*, and *spd1*, while SP1504 carried *speC* and *spd1* but lacked *ssa*. This toxin profile was reported previously in Asian *emm*1 and *emm*12 scarlet fever isolates [20,37]. The *emm*22 isolates were *speA* and *ssa* positive, but *speC* negative, which was a feature shared with *emm*22 isolates from other geographical locations, such as the UK and Europe [45,46,47,48]. The *emm*41.2 isolates SP1497 and SP1498 and *emm*75 isolate SP1506 were negative for *ssa*, *speC*, *ssa* and *spd1*, while *emm*4 isolate SP1494 was found to possess all four scarlet fever-associated toxins. The carriage of superantigen genes *speC* and *ssa* alongside DNase *spd1* is commonly reported for *emm*4 isolates, while the carriage of *speA* in this *emm* type is variable [44,47,48,49,50]. As expected, the acapsular phenotypes of all *emm*4, *emm*22, and *emm*89 isolates were accounted for by the absence of the *hasA* gene (Figure 1b). The SLO encoding gene *slo* and cysteine protease gene *speB* are highly conserved across all GAS isolates [51], and as expected, 17/17 isolates carried both genes. Macrolide-resistance gene *ermB* was also investigated, as macrolide-resistance is observed frequently in clinical scarlet fever isolates from mainland China and Hong Kong [52]. To test for potential macrolide resistance, we screened the 17 isolates for the *ermB* gene. However, none of the isolates carried the *ermB* gene (Figure 1b). Full PCR gel images can be found in Appendix A.

### 3.3. Concordance of GAS Virulence Factor Expression and Increased Virulence Factor Expression

To confirm toxin carriage and investigate toxin expression in the 17 isolates, we performed Western blot analysis to detect secreted toxins in bacterial culture supernatants grown to an OD_600_ of 0.8. Overall, the toxin profile detected by Western blot analysis matched the PCR gene screening and WGS findings (Figure 1c, Table 2). The expression of SpeA, SpeC, Spd1, and SSA was found to reflect the PCR results, except for SP1507, which was found to express SpeC but not Spd1, despite detecting *spd1* using WGS and PCR (Figure 1b). Since no mutation was detected in the *spd1* gene locus, the reason for the absence of Spd1 in SP1507 culture supernatants remains unclear. All isolates were found to secrete control proteins SLO and SpeB; however, no mature (and therefore active) SpeB was detected for SP1494, with a band detected at a higher molecular level than expected for m-SpeB. A SpeB caseinolytic assay was performed to investigate the proteolytic activity of all 17 isolates. All isolates were found to produce SpeB activity except for SP1494 and negative controls (Appendix A, Appendix A). It should be noted that SP1494 was found to possess an attenuated growth phenotype, delaying the isolate’s ability to achieve the same levels of growth as the other isolates in this study. This may explain the lack of SpeB activity detected in this assay. Full Western blot images can be found in Appendix A.

In addition, *emm*3.93 isolates SP1495, SP1496, and SP1499 all expressed high levels of SSA and SLO, while *emm*75 isolate SP1507 only displayed increased expression levels of SLO. *emm*4 isolates SP1494 and SP1508 both secreted elevated levels of SSA and SLO; however, only SP1494 expressed increased levels of SpeA. It should be noted that SP1494 was found to possess an attenuated growth phenotype. Therefore, toxin expression by this isolate should be interpreted with some caution. Intriguingly, all isolates that showed increased SLO expression also displayed increased expression levels of SSA, suggesting they may respond to similar, yet unidentified, regulatory signals.

Next, we performed an exploratory phenotypic characterisation of SLO to validate the differential expression of these toxins in the clinical isolates. SLO is a cholesterol- and glycan-dependent cytolysin that perforates the lipid bilayer of various host cells, including epithelial and immune cells [53,54,55,56]. SLO is a major GAS virulence factor, and the emergence and pathogenicity of highly virulent GAS genotypes are often associated with a high-activity promoter recombination event at the *slo* gene locus, resulting in increased SLO expression [57]. Such genotypes include numerous acapsular *emm* types, such as *emm*28, *emm*87 and *emm*89 that have all been shown to express increased levels of SLO, potentially compensating for the loss of capsule [58]. RBC lysis and haemoglobin release can be measured spectrophotometrically to quantify SLO activity [40]. Using human RBCs, the haemolytic activity of culture supernatants of all 17 isolates was compared to GAS *emm*1 strain 5448 [59] to determine SLO expression levels. Significantly increased haemolytic activity was found for *emm*3.93 (SP1495, SP1496, SP1499), *emm*4 (SP1494, SP1508) and *emm*75 (SP1507) isolates, validating the increased SLO expression levels identified by Western blotting (Figure 2).

### 3.4. Mutations Within Major GAS Virulence Factors May Drive Differential Toxin Expression

Toxin expression in GAS is a highly regulated process orchestrated by gene regulators including the control of virulence two-component regulatory system CovRS and the two stand-alone transcriptional regulators RocA and RopB [60]. Non-synonymous mutations in the coding sequences of these regulators frequently arise during infections which often leads to an altered expression of target genes, including several virulence factors [60]. We therefore hypothesised that mutations in these regulatory genes may account for differential expression levels of SpeA, SSA and SLO observed in some of the 17 clinical isolates.

Analysis of the *covRS*, *rocA* and *ropB* gene sequences revealed that both *emm*4 isolates SP1494 and SP1508 harboured a single nucleotide deletion at position 83 of *covS* resulting in an early stop codon at position 36 of the amino acid sequence and the premature truncation of CovS (Table 3). However, these isolates maintained production of the protease SpeB (Figure 1c, Appendix A), the expression of which is commonly lost in CovS truncation mutants [61]. *emm*75 isolate SP1507 has acquired an 18 base pair insertion at position 86 of the *covS* gene sequence, encoding the six amino acids IFCIFC at position 30–36 in the CovS protein sequence (Table 3). While these specific mutations have not been documented previously, inactivating mutations in CovS result in the de-repression of CovRS-regulated virulence genes such as *slo* [62,63], thus providing a possible explanation for the increased SLO expression and activity observed in these isolates. *emm*3.93 isolates SP1495, SP1496 and SP1499 were found to possess a non-synonymous SNP at position 1228 of the *rocA* gene sequence, giving rise to a premature stop codon in the RocA protein sequence (Table 3). This SNP was reportedly previously to result in an enhanced expression of SLO [64,65], which may account for an increased expression of SLO in SP1495, SP1496 and SP1499.

Furthermore, we detected a non-synonymous mutation in the SpeB regulator *ropB* in the *emm*4 isolate SP1494 with a C → T mutation at position 311 in the nucleotide sequence. This resulted in a threonine to isoleucine change at position 104 in RopB (Table 3). This T104I substitution in RopB has been shown previously to drive increased SpeB production in *emm*4 isolates [49]. However, since SP1494 showed markedly reduced expression levels of mature SpeB and displayed negligible SpeB activity (Appendix A, Appendix A), we speculate there are potentially other yet-unidentified mutations in this isolate that affect SpeB maturation.

## 4. Discussion

In this study, we conducted a molecular characterisation of 17 invasive and non-invasive clinical GAS isolates from an Australian tertiary hospital in the post-COVID period of 2021–2022, specifically focusing on non-*emm*1 GAS isolates to capture a small-scale genomic snapshot of the GAS population diversity. An increased expression of pore-forming toxin SLO was noted in isolates SP1494 and SP1508 (*emm*4), SP1495, SP1496 and SP1499 (*emm*3.93) and SP1507 (*emm*75). A further characterisation of SLO activity confirmed increased SLO activity in these isolates. This observed phenotype underscores the importance of molecular characterisation and ongoing surveillance of clinical GAS isolates.

In total, nine different *emm* types were detected in this clinical strain set. The three most prevalent *emm* types were *emm*22, *emm*12, and *emm*3.93 (three isolates each). Overall, 6/17 (35%) strains were acapsular. Capsule-negative *emm* types, such as *emm*4, *emm*22 and *emm*89, represent approximately 30% of the global GAS population and are major contributors to both invasive and non-invasive disease globally [48,66,67,68,69,70,71]. In contrast, *emm*22, *emm*12, and *emm*3.93 were the most common *emm* types; these *emm* types are common in geographical regions such as mainland China and Hong Kong or within invasive isolates from Australia [4,20,72,73,74]. The prevalence of scarlet fever-associated toxins within this strain set is of concern. Of the seven isolates found to possess *speA,* four were non-invasive isolates. While *speA* has been shown to be more commonly associated with iGAS isolates, it has also been found in association with non-invasive isolates [47,75,76,77]. It was further determined that the carriage of these toxin genes was non-*emm* type specific. 

Of the 17 isolates, only *emm4* isolate SP1494 was found to possess all tested toxins. The isolate also displayed an increased production of SLO, SpeA, and SSA in comparison to other isolates. Recent research from the United States has identified a novel fusion event between the *emm* and *enn* genes in several clinical *emm*4 isolates, resulting in the creation of a chimeric *emm* protein (designated *emm*4^C^) [78]. The authors found that the *emm*4^C^ isolates had a marked growth defect in vitro but showed increased survival in human blood and increased virulence in a murine model when compared to *emm*4 isolates [49]. The identification of the growth defect of SP1494, like that observed elsewhere [49], prompted further investigation into this isolate. It was determined that SP1494 possessed a SNP in the putative carbonic anhydrase gene *saca* responsible for the observed growth impairment as well as the chimeric *emm* gene [41]. SP1494 was also shown to possess other *emm*4^C^ clade-defining mutations, including the previously mentioned *ropB* and *saca* SNPs, confirming its position within the emergent *emm*4^C^ population, and that this hypervirulent clone has now been detected in Australia [49]. The changes in M protein sequence observed in *emm*4^C^ pose a potential difficulty to current vaccines targeting the M protein [79]. However, the development of non-M protein-based vaccines may circumvent this issue.

We also detected the presence of several mutations in key regulatory genes in this strain cohort. The invasive isolates SP1494 and SP1508, both *emm*4, and the *emm*75 isolate SP1507 carried non-synonymous mutations in *covS*. Although the repression of SpeB is a predictive marker for identifying CovS-inactivated isolates [61], none of these mutations affected SpeB expression levels. However, these isolates demonstrated high levels of SLO expression and activity, which is generally observed in CovS-inactivated isolates [61]. The *emm*3.93 isolates SP1495, SP1496 and SP1499 were all found to possess the same *rocA* mutation, which has been reported previously to contribute to *emm*3-associated GAS invasive propensity [64,65]. As mutations resulting in increased toxin production may result in the switch from colonisation to invasive disease, vaccination, which aims to prevent colonisation, would not be impacted by these mutations [80].

We wish to acknowledge that this study has several limitations. Firstly, the small size of the strain set (17 isolates) hinders the ability to make broader conclusions about *emm* type prevalence and toxin carriage in the general population. Secondly, the study is restricted to a single hospital in southeast Queensland, therefore only providing a regional snapshot of clinical GAS isolates circulating currently in Australia. Future research, incorporating larger and more diverse strain sets from multiple regions and healthcare institutions, is needed to provide a better understanding of *emm* type distribution and toxin carriage.

## 5. Conclusions

In this study, we performed the molecular characterisation of 17 non-*emm*1 clinical GAS isolates from a single tertiary hospital in southeast Queensland, Australia. The detection of the hypervirulent *emm*4^C^ clone in Australia, alongside the increased SLO production observed in several isolates, highlight the need for the continuous molecular surveillance of clinical GAS isolates. Despite our study’s limitations, these observations emphasize the importance of expanding genomic and phenotypic monitoring to detect emerging strains with increased virulence or resistance characteristics that may pose a broader public health threat.

## Figures and Tables

**Figure 1 pathogens-13-00956-f001:**
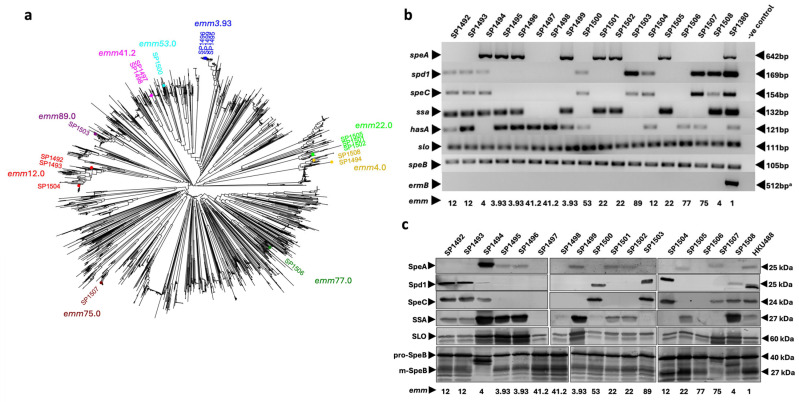
Genetic analysis and protein expression of 17 GAS clinical isolates from Gold Coast University hospital. (**a**) Population structure of 2100 globally distributed GAS genomes [36] with addition of the 17 locally acquired GAS genomes. The tree was generated using Mashtree software v1.4.6 [37]. The 17 isolates from this study are highlighted with colours matching the respective *emm* type. (**b**) PCR screening for toxin and antibiotic resistance genes. Genes examined are indicated on the left, and PCR product sizes (bp) are indicated on the right. M1_UK_ GAS isolate SP1380 was used as a positive control for all toxins [31]. ^a^ *emm*12 GAS isolate HKU16 was used as a positive control for *ermB* [23]. (**c**) Western blot analysis of toxin expression in culture supernatants. Toxins are indicated on the left, and the protein mass (kDa) is indicated on the right. M1_global_ GAS isolate HKU488 was used as a positive control. The slightly lower molecular weight band detected for SLO likely represents an SLO isoform or breakdown product [43].

**Figure 2 pathogens-13-00956-f002:**
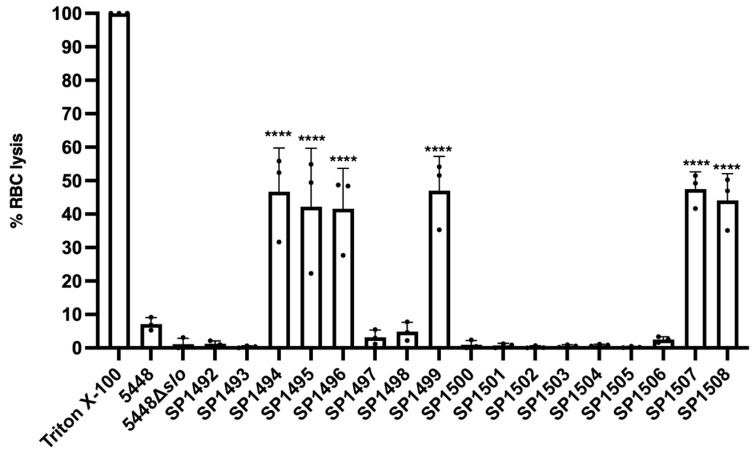
SLO haemolytic activity of 17 clinical GAS isolates from Gold Coast University Hospital. Levels of RBC lysis and the release of haemoglobin was measured spectrophotometrically at OD_405_. SLO activity was assessed in comparison to the control strain 5448. Data are plotted as the mean ± SD from three independent biological experiments, indicated as black dots. Data were analysed by one-way ANOVA and Tukey’s post-test. **** *p* < 0.0001. Positive control = 1% Triton X-100. ANOVA, analysis of variance; OD, optical density; RBCs, red blood cells; SD, standard deviation; SLO, streptolysin O.

**Table 1 pathogens-13-00956-t001:** Group A *Streptococcus* isolates from patients at Gold Coast University Hospital, 2021–2022.

GAS Isolate Designation	Age	Sex	Sample Source	Clinical Diagnosis	Non-Invasive/Invasive	*emm* Type
SP1492	13 months	M	Blood	STSS	Invasive	12
SP1493	11 months	M	Fluid lower leg	Bacteraemia	Invasive	12
SP1494	48 years	M	Thigh tissue	Bacteraemia	Invasive	4 ^a^
SP1495	17 months	M	Pleural fluid	Bacteriemia	Invasive	3.93
SP1496	15 months	F	Throat	Tonsilitis	Non-invasive	3.93
SP1497	39 years	F	Bursa fluid	Prepatellar bursitis	Invasive	41.2
SP1498	51 years	M	Blood	Bacteraemia	Invasive	41.2
SP1499	3 years	F	Swab throat	Scarlet fever	Non-invasive	3.93
SP1500	37 years	M	Blood	Bacteraemia	Invasive	53
SP1501	39 years	M	Tibia tissue	NF	Invasive	22 ^a^
SP1502	8 years	F	Throat	Scarlet fever	Non-invasive	22 ^a^
SP1503	6 years	M	Throat	Scarlet fever	Non-invasive	89 ^a^
SP1504	4 months	F	Blood	Bacteraemia	Invasive	12
SP1505	5 years	M	Throat	Scarlet fever	Non-invasive	22 ^a^
SP1506	56 years	M	Skin	NF	Invasive	77
SP1507	2 years	F	Blood	Bacteraemia	Invasive	75
SP1508	6 years	F	Blood	Bacteraemia	Invasive	4 ^a^

F, female; M, male; NF, necrotising fasciitis; STSS, streptococcal toxic shock syndrome. ^a^ Acapsular *emm* types.

**Table 2 pathogens-13-00956-t002:** Summary of all identified toxins in the 17 clinical GAS isolates analysed in this study.

Isolate	*emm* Type	*speA* ^a^	*speC* ^a^	*ssa* ^a^	*spd1* ^a^	*hasA* ^b^	*speB* ^a^	*slo* ^a^	*ermB* ^b^
SP1492	12	−	+	+	+	+	+	+	−
SP1493	12	−	+	+	+	+	+	+	−
SP1494	4	+	+	+	+	−	+	+	−
SP1495	3.93	+	−	+	−	+	+	+	−
SP1496	3.93	+	−	+	−	+	+	+	−
SP1497	41.2	−	−	−	−	+	+	+	−
SP1498	41.2	−	−	−	−	+	+	+	−
SP1499	3.93	+	−	+	−	+	+	+	−
SP1500	53	−	+	−	+	+	+	+	−
SP1501	22	+	−	+	−	−	+	+	−
SP1502	22	+	−	+	−	−	+	+	−
SP1503	89	−	+	−	+	−	+	+	−
SP1504	12	-	+	−	+	+	+	+	−
SP1505	22	+	−	+	−	−	+	+	−
SP1506	77	−	−	−	−	+	+	+	−
SP1507	75	−	+	−	Ind ^c^	+	+	+	−
SP1508	4	−	+	+	+	−	+	+	−

GAS, Group A *Streptococcus*; PCR, polymerase chain reaction; WGS, whole genome sequencing; +, positive; −, negative, Ind, indefinite. ^a^ identified using WGS, PCR and Western blot analysis. ^b^ identified using WGS and PCR analysis only.^c^
*spd1* gene positive but Spd1 expression negative.

**Table 3 pathogens-13-00956-t003:** GAS master regulators and associated non-synonymous mutations of the 17 clinical GAS isolates analysed in this study.

Isolate	*covR*		*covS*		*rocA*		*ropB*	
	Carriage	Mutation	Carriage	Mutation	Carriage	Mutation	Carriage	Mutation
SP1492	+	−	+	−	+	−	+	−
SP1493	+	−	+	−	+	−	+	−
SP1494	+	−	+	truncation	+	−	+	T104I ^a^
SP1495	+	−	+	−	+	insertion	+	−
SP1496	+	−	+	−	+	insertion	+	−
SP1497	+	−	+	−	+	−	+	−
SP1498	+	−	+	−	+	−	+	−
SP1499	+	−	+	−	+	insertion	+	−
SP1500	+	−	+	−	+	−	+	−
SP1501	+	−	+	−	+	−	+	−
SP1502	+	−	+	−	+	−	+	−
SP1503	+	−	+	−	+	−	+	−
SP1504	+	−	+	−	+	−	+	−
SP1505	+	−	+	−	+	−	+	−
SP1506	+	−	+	−	+	−	+	−
SP1507	+	−	+	insertion	+	−	+	−
SP1508	+	−	+	truncation	+	−	+	−

GAS, Group A *Streptococcus*; +, gene detected; −, mutation not detected. ^a^ Threonine substituted for isoleucine at position 104 of the amino acid sequence.

## Data Availability

Whole genome sequencing data obtained in this study were submitted to the Sequence Read Archive database (https://www.ncbi.nlm.nih.gov/sra (accessed on 30 September 2024); BioProject accession PRJNA1172375).

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
