# Peer review of "Clinical Snapshot of Group A Streptococcal Isolates from an Australian Tertiary Hospital"

_pathogens, 2024, doi:10.3390/pathogens13110956_

Round 1
Reviewer 1 Report
Comments and Suggestions for Authors
Title
The title is clear and accurately describes the main content of the study, indicating the focus on "Group A streptococcus isolates" and the specific location.
Abstract
The abstract provides a concise overview of the study, its objectives, the methodology (molecular characterization), and the main results, making it easy to understand the article’s main points. Despite the word limit in the abstract, I suggest explicitly mentioning the importance of the findings in clinical and epidemiological terms so that readers can quickly grasp the study's contribution to medical practice or outbreak control.
Introduction
The introduction does a good job of contextualizing the problem, addressing the main diseases caused by Streptococcus pyogenes and the growing concern over invasive infections and global outbreaks. Relevant data are presented. I suggest simplifying the introduction by removing secondary information and focusing on the specific gap the study aims to address. This would make the argument more direct and effective.
Methodology
The study employs advanced molecular biology techniques (such as whole-genome sequencing and PCR) and provides sufficient details about bacterial growth conditions and genetic analyses, which ensures reproducibility. It is well-written and clear, allowing for easy understanding.
Results
The results are detailed and well-presented, with figures and tables that facilitate data visualization, such as the phylogenetic tree and toxin expression analyses.
If possible, I suggest simplifying the presentation of the main results by highlighting the most relevant findings before diving into the technical details.
Discussion
The discussion is solid and makes a good connection between the results and the existing literature, particularly regarding the virulence of the emm types and the absence of capsules. The study highlights the importance of continuous molecular surveillance, which is an important and relevant point given the emergence of virulent strains.
I suggest addressing the study's limitations (which are mentioned) in a more pragmatic way, especially regarding the limited sampling (number of isolates) and the restricted geographic impact. The discussion could also benefit from more reflection on the potential impact of the findings in terms of vaccine development or new therapies.
Conclusion
I suggest creating a standalone conclusion section, possibly using text from the final paragraph, to summarize the study.
Line 184 – highlighted in yellow for some reason?
Reviewer 2 Report
Comments and Suggestions for Authors
This manuscript presents methodological rigor in all its sections. The results are well described and contribute to the scientific literature. However, the authors could include the patient's clinical evolution in Table 1. Did the patients die or survive?
Reviewer 3 Report
Comments and Suggestions for Authors
Comments and suggestions to the authors:
1. Please provide ethical information for the clinical and in vitro studies.
2. In the discussion section, the author noted that the clinical samples were collected during 2021-2022, following the COVID-19 pandemic. Please discuss whether there are any significant differences between these samples and those collected before the pandemic as a result of public health interventions.
3. In Table 3, SP1942, SP1943, and SP1505 are all emm12 isolates; however, there are significant differences in the expression of ssa. Please clarify the reasons for these differences.
4. Given that Queensland has high rates of rheumatic fever and rheumatic heart disease, the prevention and control of GAS infections are essential for improving the health of local residents. Please provide a brief comment in the manuscript on the impact of these mutations and virulence changes on treatment and the development of a region-specific GAS vaccine.
Round 2
Reviewer 1 Report
Comments and Suggestions for Authors
The authors made an effort to respond positively to most of the suggestions I made. On rereading the article, it seems to me to be more structured, with more content and in a way that is easier to read and understand. I have no further suggestions. I suggest a new reading, to mitigate any necessary details and flaws
Author Response
Comment 1: The authors made an effort to respond positively to most of the suggestions I made. On rereading the article, it seems to me to be more structured, with more content and in a way that is easier to read and understand. I have no further suggestions. I suggest a new reading, to mitigate any necessary details and flaws.
Response 1: We would like to thank the reviewer again for their positive feedback. We agree that the article is now easier to read, avoiding any unnecessary details and flaws.